# Analysis of the Structural Dynamics of Proteins in the Ligand-Unbound and -Bound States by Diffracted X-ray Tracking

**DOI:** 10.3390/ijms241813717

**Published:** 2023-09-06

**Authors:** Masayuki Oda

**Affiliations:** Graduate School of Life and Environmental Sciences, Kyoto Prefectural University, 1-5 Hangi-cho, Shimogamo, Sakyo-ku, Kyoto 606-8522, Japan; oda@kpu.ac.jp

**Keywords:** antibody, binding energy, DNA-binding protein, fluctuation, helix-bundle protein

## Abstract

Although many protein structures have been determined at atomic resolution, the majority of them are static and represent only the most stable or averaged structures in solution. When a protein binds to its ligand, it usually undergoes fluctuation and changes its conformation. One attractive method for obtaining an accurate view of proteins in solution, which is required for applications such as the rational design of proteins and structure-based drug design, is diffracted X-ray tracking (DXT). DXT can detect the protein structural dynamics on a timeline via gold nanocrystals attached to the protein. Here, the structure dynamics of single-chain Fv antibodies, helix bundle-forming de novo designed proteins, and DNA-binding proteins in both ligand-unbound and ligand-bound states were analyzed using the DXT method. The resultant mean square angular displacements (MSD) curves in both the tilting and twisting directions clearly demonstrated that structural fluctuations were suppressed upon ligand binding, and the binding energies determined using the angular diffusion coefficients from the MSD agreed well with the binding thermodynamics determined using isothermal titration calorimetry. In addition, the size of gold nanocrystals is discussed, which is one of the technical concerns of DXT.

## 1. Introduction

Proteins in solution fluctuate to varying degrees and time scales. The most stable or averaged structures can be determined at high resolution using conventional methods, such as X-ray crystallography and nuclear magnetic resonance (NMR) [1,2,3,4]. NMR relaxation and hydrogen-deuterium exchange experiments can be used to study protein structural dynamics [5,6,7], but the results are an ensemble average on a limited time scale. Entropy change, which includes the contribution of protein structural dynamics, can be determined using calorimetric measurements [8,9,10,11]. Differential scanning calorimetry (DSC) and isothermal titration calorimetry (ITC) can be used to detect folding and binding thermodynamics, respectively, both of which are derived from total thermodynamics in solution. Assuming that a protein contains the same amino acids, it will mostly fold into a stable tertiary structure, but the respective molecules in solution will have subtly different conformations at any given time and change the conformation in a time-dependent manner. At low energy levels, the subtly different structures of a protein coexist and are exchangeable with each other [12,13,14]. Protein function is closely related to dynamic protein structures. To observe the real view of proteins, innovative methods for detecting the structural dynamics at the single molecular level in a time-dependent manner are required. Sasaki et al. succeeded in time-resolved X-ray observations of dynamical motions of individual functional proteins and DNA in aqueous solutions for the first time in the world [15,16,17]. This single molecular detection system was termed diffracted X-ray tracking (DXT) and became a pioneering method for determining protein motion. Protein structural dynamics can be determined using a brilliant light source and a high-speed detector to detect the time-dependent movement of a gold nanocrystal attached to the target protein in real-time, in the range of μsec to msec. Using DXT, the structural dynamics of various proteins and those in ligand–protein interactions have been analyzed [18,19,20,21,22,23,24,25,26,27,28]. We used DXT to analyze the change in structural dynamics of relatively smaller globular proteins, a single-chain variable fragment (scFv) antibody (26 k), a de novo designed protein (8 k), and a DNA-binding protein (12 k), as they interacted with other molecules [29,30,31,32]. When a protein binds to another molecule, various types of conformational changes are observed, which are closely correlated with subsequent events such as signal transduction and transcriptional regulation. Antibodies or immunoglobulins interact with antigens of different shapes and sizes, and the interactions have been described as the protein recognition modes such as lock and key, indued fit, and population shift [33,34]. The antigen binding to B cell receptors (BCRs) composed of membrane immunoglobulin and Igα/Igβ results in the transduction of signals to the cell interior. The signal transduction would relate to the allosteric conformational change in the antibody constant region upon antigen binding [35], especially for monovalent antigens, which could not induce BCR crosslinking or aggregation. Although many crystal structures of liganded and unliganded Fab or Fv fragments have been determined, the central question about the relationship between conformational change and signal transduction remains unclear, mainly because of the little information on the structural dynamics of antibodies. To observe protein structural dynamics similar to those without immobilization, the protein was immobilized on the DXT substrate via an N-terminal polyhistidine tag, followed by attachment of the gold nanocrystal via the sulfur atom of a Met residue. The angular displacement of the nanocrystal used as a motion tracer in the DXT measurements was analyzed along two rotational axes: tilting (θ) and twisting (λ) [7]. Under each condition, the trajectories of the nanocrystal motions were tracked, and the traced data were used to calculate the mean square angular displacement (MSD) curves. The angular diffusion coefficients (*D*) calculated from the slope of the MSD curves were used to evaluate protein motion. Furthermore, the difference in *D* values was used to evaluate the motion change caused by binding to another molecule. The binding energies were calculated from the *D* values using the following Equation:Δ*G* = α ln(*D*_b_/*D*_f_) + β(1)
where *D*_f_ and *D*_b_ represent the free and bound states of the proteins, respectively. Δ*G* values were calculated using the empirical values of α (14.6 kJ/mol) and β (−30.1 kJ/mol) [36]. These values were compared with the Δ*G* values determined from solution binding experiments. Because the proteins we studied using DXT were smaller than the gold nanocrystals with a diameter of 40–80 nm, multiple protein molecules could attach to a single nanocrystal. Regarding the technical concerns of DXT, the effect of immobilization density of the protein is also discussed.

## 2. Antibody in the Antigen-Unbound and -Bound States

It was previously shown that at least two types of antibodies are secreted after immunization with (4-hydroxy-3-nitrophenyl)acetyl (NP); one has Tyr, and another has Gly at position 95 of the heavy chain (referred to as Tyr95- and Gly95-type) [37]. The former appeared at an early stage, while the latter appeared at a late stage, i.e., after secondary immunization, although Fv domains of these antibodies were encoded by the same genes of variable heavy and light chains [37]. Primary antibodies produced at an early stage of immunization are characterized by low antigen affinity, while those secreted at a late stage possess a higher affinity, which is referred to as affinity maturation and which has been shown to be induced by somatic hypermutation [38,39,40,41]. The NP-binding affinity of C6, an affinity maturated and Gly95-type antibody, at 25 °C determined using ITC was 3.3 × 10^7^ M^−1,^ which was approximately 100-times higher than that of the germline-type antibody [37]. ITC detects the heat generated by biomolecular interactions, allowing for the accurate determination of binding thermodynamics [42]. A single-chain Fv (scFv) of C6 in which the variable light chain was linked to the variable heavy chain via a flexible (G_4_S)_3_ linker, comprising three repeats of the Gly–Gly–Gly–Gly–Ser peptide, was generated and overexpressed in *Escherichia coli* [29]. Since the protein was expressed in the insoluble fraction, it was solubilized using urea and refolded using a stepwise dilution method. The scFv with the N-terminal polyhistidine tag was purified using an antigen column, followed by further purification using size exclusion chromatography (SEC). Because the scFv fraction bound to an antigen column also included the aggregated or multimeric form, the SEC method to obtain the scFv monomer would be effective. To evaluate the correct folding of the scFv monomer, the binding stoichiometry determined using ITC would be one of the most reliable information. When the scFv refolds correctly, it could bind to its antigen in a 1:1 ratio. Purified C6 scFv bound to NP with an affinity of 5.68 × 10^7^ M^−1^, similar to its parent monoclonal antibody [43], supporting the notion that the scFv was correctly folded. The crystal structure of C6 scFv in complex with NP was determined at a 1.65 Å atomic resolution (Figure 1) and revealed the antigen recognition mechanism, primarily via the residues in the complementarity determining region (CDR) [44]. The somatic hypermutation occurs at 17 residues in C6. Among these, Lys58HArg in CDR2 of the heavy chain is the only mutation involved directly in the interaction with NP through the formation of hydrogen bonds. Since this mutation is observed widely in Gly95H-type antibodies, it is referred to as a parallel mutation and was expected to be a common strategy to raise the affinity of this type of antibody. The hydroxyl group of NP is recognized by the hydrogen bond with Arg50H and Arg58H in C6. This can explain the unique property of anti-NP antibodies, preferential binding to the phenolate form of NP than to the phenolic form [45].

The ability to modify specific sites through site-directed mutagenesis is one advantage of producing scFv antibodies. To label the gold nanocrystals, the Asn54 residue of the C6 heavy chain was replaced with Met (Figure 1). Because the residue is located on the CDR2 loop and the side-chain is exposed to the solvent, antigen binding would have a significant impact on movement. DXT experiments for C6 scFv in the absence or presence of the antigen in 10 mM HEPES (pH 7.3) containing 140 mM NaCl were carried out, revealing that the *D* values for both the θ and λ directions decreased upon antigen binding (Table 1). These results indicate that antigen binding inhibited the motion of C6 scFv. The calculated Δ*G* values using Equation (1) were −41.5 and −43.5 kJ/mol for θ and λ directions, respectively. These values agree well with the ITC value of −44.2 kJ/mol [43], suggesting that the fluctuations detected at the single-molecule level are well reflected by the antigen binding event in solution. The contribution of the decreased C6 scFv fluctuation upon antigen binding was included in the negative value of the binding entropy change determined using ITC [43]. This decreased fluctuation is also supported by the increased stability of C6 scFv upon antigen binding. The thermodynamic analysis of the binding and folding of antibodies during affinity maturation revealed a trade-off between the binding affinity and thermostability of antibodies in an antigen-unbound state [46]. A germline-type antibody with low antigen-binding affinity would have a relatively rigid structure, whereas a maturated-type antibody would have increased structural flexibility to bind to a specific antigen with high affinity.

## 3. De Novo Designed Protein in the Metal-Unbound and -Bound States

In the presence of metal ions coordinated to His or Cys residues, model proteins were designed to change their structure from a random coil to a helix-bundle by forming a stable hydrophobic core [47,48,49]. The α-helical coiled-coil has a representative amino acid sequence of (abcdefg)_n_ heptad repeats, in which the residues at positions a and d face the hydrophobic core. A de novo designed three-helix-bundle protein, α3D, consists of 72 residues having three α-helical portions between 3–20 (helix 1), 27–44 (helix 2), and 52–69 (helix 3), and its tertiary structure was determined by NMR [50]. The hydrophobic residues (9-residues) of α3D were changed to His (6-residues) and other amino acids (3-residues) (Figure 2), resulting in structure destabilizing and refolding into the helix-bundle by metal ion binding to His residues [30]. The conformational changes induced by metal ion binding were analyzed using circular dichroism (CD), showing that the destabilization effects largely depended on the substituted amino acids in the 3-residues. Of all the peptides designed and purified using Ni-nitrilotriacetic acid and reverse-phase HPLC columns, HA, in which the 3-residues are Ala (Figure 2A), showed distinct conformational changes induced by the binding of the metal ion; the helical content increased upon binding to Zn^2+^ at most. ITC experiments showed that the Zn^2+^ binding affinity of HA at 25 °C was 2.06 × 10^6^ M^−1^ [51]. To label with gold nanocrystals, a Met residue was introduced at residue 64, which was located on the third helix and exposed to the solvent (Figure 2). When HA folded into a helix-bundle structure, the position of the gold nanocrystal was closer to the substrate because the N-terminal His-tag of HA was attached to the substrate. DXT experiments were carried out on HA in the absence or presence of Zn^2+^ in 20 mM PIPES (pH 7.0) containing 100 mM NaCl. The slopes of the MSD curves, *D* values, differed markedly between the Zn^2+^-free and Zn^2+^-bound forms of HA in both the θ and λ directions (Table 1). The lower *D* values upon Zn^2+^ binding indicated that the large-scale motion of HA was suppressed, mainly because of the formation of the three helix-bundle structures. The Δ*G* values calculated using Equation (1) were −35.3 and −36.8 kJ/mol for θ and λ directions, respectively, which were comparable to the −36.0 kJ/mol determined by the ITC experiments.

The Ala residues in HA were changed to Val, Ile, and Leu to produce the HA-related proteins HV, HI, and HL, respectively. CD and ITC were used to analyze the structural and Zn^2+^-binding properties [51]. Far-UV CD spectra indicated that HV and HI changed their structures from random coil-like to α-helix-rich following Zn^2+^ binding, similar to that observed for HA. In contrast, HL formed an α-helix-rich structure even in the absence of Zn^2+^. Notably, the near-UV CD spectrum of HI was similar to that of HL in the presence of Zn^2+^, different from those of HA and HV. ITC experiments showed that the Zn^2+^ binding affinity of HL was the highest, while that of HV was the lowest. Taken together, the structural dynamics of these peptides would be largely influenced by the hydrophobic core residues and would correlate with the binding thermodynamics of the metal ions. ESR was also used to investigate the metal ion coordination of Cu^2+^, instead of Zn^2+^, to HA, HV, HI, and HL. The results revealed that the proteins in either metal ion-bound or -unbound states had distinct properties, such as random, helix-bundle, and partially folded structures. The ratios of random and helix-bundle structures would change when the metal ions bound to HA, but both structures would still coexist in equilibrium. ESR analysis also confirmed that HA in the metal ion-bound state fluctuated significantly, similar to HV but different from HI and HL. Among the four proteins, HL in the metal ion-bound state was the most rigid, with the least fluctuation. These proteins are simple models in which the hydrophobic core significantly influences the protein fluctuations. In the next step, single-molecule analysis will provide additional information. For example, the population of the respective structures will be equilibrated, and the change in the metal ion binding will be revealed.

## 4. DNA-Binding Protein in the DNA-Unbound and -Bound States

c-Myb is a transcriptional activator that binds to the specific DNA sequence PyAAC(G/T)G, where Py represents a pyrimidine [52,53,54]. The DNA-binding domain consists of three imperfect 51 or 52 residue repeats (designated R1, R2 and R3 from the N-terminus) [55], with the last two repeats, R2 and R3, being the minimum unit required for specific DNA-binding [56]. NMR was used to determine the complex structure of R2R3 and DNA (Figure 3), which revealed that both R2 and R3 were composed of three helices with a helix-turn-helix variation motif, and each third helix was engaged in specific base recognition [57]. For DXT experiments, c-Myb R2R3 with an N-terminal polyhistidine tag was overexpressed in *Escherichia coli* and purified using the Ni-nitrilotriacetic acid column, followed by SEC. To attach a gold nanocrystal to a specific site of R2R3, a Met residue was introduced via mutation into the 135th residue located on the third helix of R2, and residues Cys130 and Met189 were changed to Ile and Ala, respectively. DXT could be used to analyze the time-dependent movement of a gold nanocrystal attached to Met135 of R2R3 because the C130I/H135M/M189A mutant only has a sulfur-containing residue at 135th [31]. The binding affinity of this mutant to the 22-mer DNA containing AACTG base sequence in PBS (10 mM phosphate buffer, 140 mM NaCl) was determined using ITC and was found to be 2.27 × 10^7^ M^−1^ at 20 °C, similar to those of R2R3 wild-type and C130I mutant [58]. DXT experiments in PBS for the C130I/H135M/M189A mutant in both the DNA-bound and DNA-unbound states revealed that the *D* values for both the θ and λ directions decreased upon DNA-binding (Table 1). The Δ*G* values calculated using Equation (1) were −44.2 and −41.0 kJ/mol for the θ and λ directions, respectively, which were comparable to the −41.3 kJ/mol determined using ITC.

The DNA-binding affinity of R2R3 decreased as the ionic strength of the solvent increased [58], mainly because the high ionic strength weakened the electrostatic interactions between DNA and protein. This is supported by the DXT results, which showed that as salt concentration increased, the difference in *D* values between the DNA-bound and DNA-unbound states decreased (Table 1). The *D* value in the DNA-unbound state decreased significantly at a high salt concentration (PBS containing 250 mM NaCl), indicating that R2R3 fluctuated more in PBS than in PBS containing 250 mM NaCl. This would be due to fewer hydrophobic contacts in the core region and greater electrostatic repulsion on the surface of R2R3, which contains abundant basic amino acid residues, as is typical for DNA-binding proteins. The cavity in the R2 hydrophobic core is critical for the DNA-binding function of R2R3 [58,59] and contributes to its fluctuation. R2 would destabilize and increase its fluctuation if there were less hydrophobic contacts. Despite the increased electrostatic interaction with DNA, a more fluctuating structure at low ionic strength would lose its ability to bind DNA [60]. Similar to the results of the de novo designed proteins described above, protein folding derived primarily from hydrophobic contacts in the core region would largely influence protein fluctuation, which would closely correlate with function. R2R3 fluctuates properly to express its DNA-binding ability under physiological conditions. These are also supported by the previous reports showing that c-Myb R2R3 is like a “semi-intrinsically disordered” protein and changes the structure into a well-folded form upon DNA-binding [60]. The calorimetric enthalpy change of R2R3 in the DNA-unbound state was much smaller than that in the DNA-bound state, strongly indicating that R2R3 alone fluctuated largely in solution, resulting in weak intramolecular interactions. Intrinsically disordered proteins have been found in transcription factors [61,62,63], and their ultimate fluctuation plays a pivotal role in their binding to the specific site on DNA.

## 5. Conformational Entropy

ITC detects the binding enthalpy change in biomolecular interactions directly from binding heat, thereby enabling the determination of the binding entropy change more accurately than other methods. Table 2 summarizes the binding thermodynamics for C6 scFv and NP, HA and Zn^2+^, and c-Myb R2R3 and DNA, according to the ITC. All binding was derived from favorable enthalpy changes, partially compensated by unfavorable entropy changes. The entropy change included the contribution from the conformational entropy upon binding. In most biomolecular interactions, conformational entropy decreases upon binding to another molecule, resulting in a negative value for entropy change. In contrast, dehydration effects would contribute to the positive value of entropy change, as observed in hydrophobic interactions [64]. When the hydrophobic residues are exposed to the solvent, water molecules interact with the residues. Then, the hydrophobic residues interact with another molecule, and the ordered water molecules are released into the solvent, resulting in increased entropy. The ITC determined an entropy change, which included conformational entropies from all molecules in the solution.

It should be noted that the entropy change in HA and Zn^2+^ was significantly more negative than others (Table 2). This would be caused by the significant conformational change of HA upon Zn^2+^ binding. In the metal-free state, HA exists in a random coil-like structure. The conformational entropy in HA significantly decreased in the metal-bound state because HA changed into a helix-rich structure. The conformational change was induced upon antigen binding to C6 scFv, but the degree was much lower. In the comparison of DNA-binding to c-Myb R2R3 under different salt concentrations, the binding entropy change in 140 mM NaCl was more negative than that in 250 mM NaCl. The difference could also be caused by conformational entropy. Our previous study showed that the stability of R2R3 increased with increasing NaCl concentrations [60]. The decreased fluctuations in the high ionic strength observed in DXT experiments correlate well with increased stability. The excess positive charges in R2R3 likely contribute to the long-range electrostatic interaction with DNA and to the instability of R2R3 when present in the DNA-unbound state, owing to charge repulsion. Lowered electrostatic effects under high ionic conditions enhanced the contribution of hydrophobic interactions, especially in the hydrophobic core of R2 and R3, resulting in decreased fluctuations.

## 6. Concluding Remarks and Future Perspectives

DXT experiments on globular and relatively smaller proteins, a scFv antibody (26 k), a helix-bundle protein (8 k), and c-Myb R2R3 (12 k), revealed that their structural motions decreased upon binding to an antigen, metal ion, and DNA, respectively. The *D* values calculated from the slopes of the MSD curves could explain the respective binding energies in the solution, indicating that the fluctuations were well reflected by the binding event in the solution. Because gold nanocrystals with diameters of 40–80 nm are larger than the proteins with a diameter of 5–10 nm discussed in this review, multiple proteins may bind to a single gold nanocrystal. DXT experiments were performed using the c-Myb R2R3 C130I/H135M/M189A mutant at different protein immobilization densities [32]. In the absence of DNA, no differences were observed between the two immobilization densities. In contrast, in the presence of DNA, more dynamically moving species were observed at low immobilization densities, in addition to species similar to those observed at high immobilization densities. The histograms of the angular displacement corresponding to θ direction analyzed for each 100 msec provided two histogram peaks at low immobilization density, whereas only one peak was observed at high immobilization density (Figure 4). The phenomena may be caused by the attachment of multiple proteins to a single gold nanocrystal. In future DXT experiments, the smaller size of the gold nanocrystal will enable analysis of the structural dynamics at the complete single-molecule level, even for small globular proteins, and detection of motion at a higher resolution. Using the smaller nanocrystals, it would be hard to obtain a signal from them. This is one of the critical issues in improving the DXT methods, especially when applied to small proteins. The high spatial resolution of structural motion, similar to the static structural data of proteins, will reveal new insights into the world of proteins. For small proteins, because structural dynamics could be well-determined using NMR, such as relaxation and hydrogen–deuterium exchange experiments [5,6,7], the comparable analyses using both DXT and NMR will provide new insights on the protein structural dynamics and the change upon the ligand binding.

## Figures and Tables

**Figure 1 ijms-24-13717-f001:**
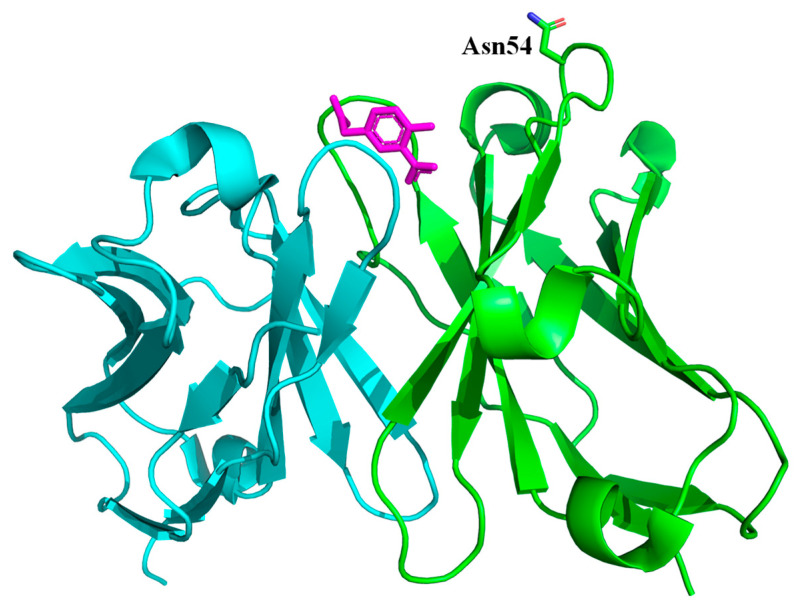
Crystal structure of C6 scFv in complex with NP (PDB code; 6K4Z). The light and heavy chains of C6 scFv are indicated in cyan and green, respectively. The antigen, NP (magenta), and the side-chain of the 54th residue (Asn) of the heavy chain are indicated by the stick-model. The figure was drawn by PyMOL 2.5.2 (Schrodinger, LLC, New York, NY, USA).

**Figure 2 ijms-24-13717-f002:**
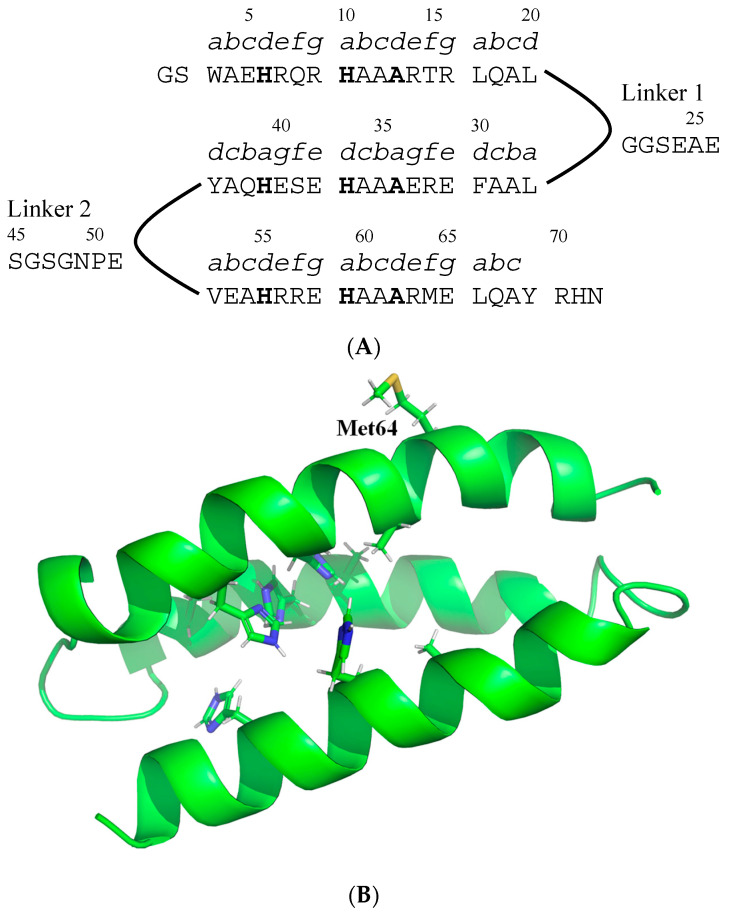
(**A**) Amino acid sequence of HA [30]. The His residues involved in metal ion binding and the Ala residues changed from hydrophobic amino acids of α3D are indicated in bold. The positions a—g are also indicated above the amino acids. (**B**) Structure model of HA based on NMR structure of α3D (PDB code; 2A3D). The side-chains of His and Ala in the hydrophobic core and that of Met64 are indicated by the stick-model. The figure was drawn by PyMOL 2.5.2 (Schrodinger, LLC, New York, NY, USA).

**Figure 3 ijms-24-13717-f003:**
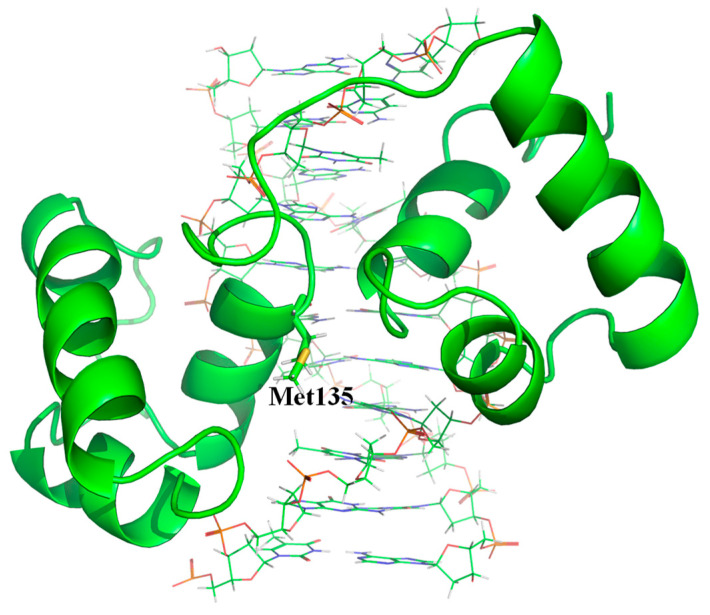
NMR structure of c-Myb R2R3 in complex with DNA (PDB code; 1MSE). The residue His135 is mutated to Met and its side-chain is indicated by stick-model. The figure was drawn by PyMOL 2.5.2 (Schrodinger, LLC, New York, NY, USA).

**Figure 4 ijms-24-13717-f004:**
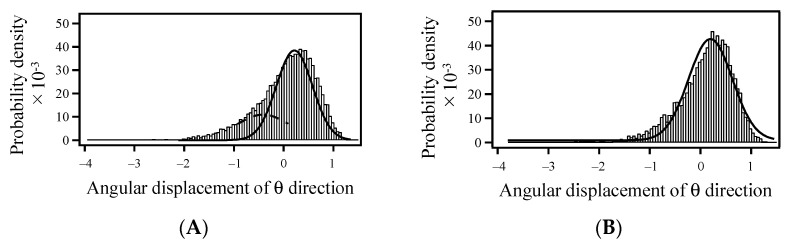
Histograms of the Δ absolute angular displacement of gold nanocrystal on R2R3 mutant C130I/H135M/M189A in θ direction within 100 ms. The histograms were compared in the presence of DNA at low (**A**) and high (**B**) immobilization densities. The figures were slightly modified from Figure 1A,B in [32].

**Table 1 ijms-24-13717-t001:** Angular diffusion coefficient, D (rad^2^/s).

	Tilting (θ)	Twisting (λ)
C6 scFv ^a^	2.04 (±0.03) × 10^−5^	8.65 (±0.12) × 10^−6^
C6 scFv + antigen (NP) ^a^	9.34 (±0.07) × 10^−6^	3.46 (±0.09) × 10^−6^
HA ^b^	1.08 (± 0.03) × 10^−5^	6.64 (± 0.09) × 10^−6^
HA + Zn^2+ b^	7.57 (± 0.15) × 10^−6^	4.19 (± 0.03) × 10^−6^
R2R3 in PBS, 140 mM NaCl ^c^	1.31 (±0.04) × 10^−5^	2.20 (±0.39) × 10^−6^
R2R3 + DNA in PBS, 140 mM NaCl ^c^	4.99 (±0.07) × 10^−6^	1.04 (±0.07) × 10^−6^
R2R3 in PBS, 250 mM NaCl ^c^	7.01 (±0.25) × 10^−6^	1.52 (±0.08) × 10^−6^
R2R3 + DNA in PBS, 250 mM NaCl ^c^	5.66 (±0.25) × 10^−6^	1.95 (±0.09) × 10^−6^

^a^ Data were taken from Sato et al., 2016 [29]. ^b^ Data were taken from Usui et al., 2017 [30]. ^c^ Data were taken from Hosoe et al., 2018 [31].

**Table 2 ijms-24-13717-t002:** Binding thermodynamics.

	Δ*G* (kJ/mol)	Δ*H* (kJ/mol)	*T*Δ*S* (kJ/mol)
C6 scFv + antigen (NP) ^a^	–44.2	–52.1	–7.9
HA + Zn^2+ b^	–36.0	–71.2	–35.2
R2R3 + DNA in PBS, 140 mM NaCl ^c^	–41.3	–48.8	–7.5
R2R3 + DNA in PBS, 250 mM NaCl ^c^	–36.8	–38.1	–1.3

^a^ Data were taken from Sato et al., 2017 [43]. ^b^ Data were taken from Tanaka et al., 2021 [51]. ^c^ Data were taken from Hosoe et al., 2018 [31].

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
