# Peer review of "Analysis of the Structural Dynamics of Proteins in the Ligand-Unbound and -Bound States by Diffracted X-ray Tracking"

_ijms, 2023, doi:10.3390/ijms241813717_

Round 1
Reviewer 1 Report
The review Analysis of the structural dynamics of proteins in the ligand-unbound and -bound states by diffracted X-ray tracking by Masayuki Oda focuses on the “dynamics/motion” of proteins detected by diffracted X-ray tracking -DXT. The DXT is mainly worked by Japanese groups due to the required equipment. Most of the experiments may be complemented by ITC/DSC techniques ( or other) . In general the manuscript is readable although the organization /structuration is a bit problematic. I believe that the Review needs to be reworked to improve the connectivity and probably some additions of papers, figures, schemes etc.. In the present state I could not recommend it for publication.
Regarding the introduction, It would be nice to have an introduction of the DXT basics (what is needed design in mind etc. ) or alternatively point out to a reference e.g. there is a very nice review on DXT in IJMS “Measuring Intramolecular Dynamics of Membrane Proteins. Int. J. Mol. Sci. that explains the basics of the method. 2022, 23, 2343. “
Secondly of this is a review I would recommend to include more protein-ligand DXT papers below are some suggestions :
· Mio, Kazuhiro, et al. "Living-cell diffracted X-ray tracking analysis confirmed internal salt bridge is critical for ligand-induced twisting motion of serotonin receptors." International Journal of Molecular Sciences 22.10 (2021): 5285.
· Oishi, Koichiro, et al. "Ligand-Dependent Intramolecular Motion of Native Nicotinic Acetylcholine Receptors Determined in Living Myotube Cells via Diffracted X-ray Tracking." International Journal of Molecular Sciences 24.15 (2023): 12069.
· Mio, Kazuhiro, et al. "Real-Time Observation of Capsaicin-Induced Intracellular Domain Dynamics of TRPV1 Using the Diffracted X-ray Tracking Method." Membranes 13.8 (2023): 708.
· Kuramochi, Masahiro, et al. "Dynamic motions of ice-binding proteins in living Caenorhabditis elegans using diffracted X-ray blinking and tracking." Biochemistry and Biophysics Reports 29 (2022): 101224.
· Sasaki, Daisuke, Tatsuya Arai, Yue Yang, Masahiro Kuramochi, Wakako Furuyama, Asuka Nanbo, Hiroshi Sekiguchi, Nobuhiro Morone, Kazuhiro Mio, and Yuji C. Sasaki. "Micro-second Time-resolved X-ray Single-molecule Internal Motions of SARS-CoV-2 Spike Variants." (2022).
· Yang, Yue, Tatsuya Arai, DAISUKE SASAKI, Hidetoshi Inagaki, Sumiko Ohashi, Masahiro Kuramochi, Hiroshi Sekiguchi, Kazuhiro Mio, Tai Kubo, and Yuji Sasaki. "The twisting direction of nAChR α7-ivermectin is opposite to that of the activated state." (2022).
The four points ( introduction is removed) listed below seem to be unrelated and the transition is quite unforeseen :
2. Antibody in the antigen-unbound and -bound states
3. De novo designed protein in the metal-unbound and -bound states
4. DNA-binding protein in the DNA-unbound and -bound states
5. Conformational entropy
The claimed in the title analysis MSD or size of Au nanocrystals is rather ephemeral.
Some of the figures have been previously published e,g, Figure 2a = Fig.1 from https://doi.org/10.1016/j.bpc.2021.106661 just the underline removed. citation missing in the captions , …
Minor
A lot of abbreviations have been introduced without
Fv domains , Fv antibodies Fv antibodies
Bcr
Line 82 It was previously shown.. please add reference(s) ..
What is C6?
(G4S)3 linker
HA line 155 ?
Author Response
Thank you very much for your valuable comments.
Comment 1: Regarding the introduction, It would be nice to have an introduction of the DXT basics (what is needed design in mind etc.) or alternatively point out to a reference e.g. there is a very nice review on DXT in IJMS “Measuring Intramolecular Dynamics of Membrane Proteins. Int. J. Mol. Sci. that explains the basics of the method. 2022, 23, 2343. “
Reply 1: In lines 43 and 69, I have added the reference [7].
Comment 2: Secondly of this is a review I would recommend to include more protein-ligand DXT papers below are some suggestions:
Reply 2: I have added the sentence in liens 47-48 “Using DXT, structural dynamics in ligand-protein interactions have been analyzed [8-11]”.
Comment 3: The four points (introduction is removed) listed below seem to be unrelated and the transition is quite unforeseen.
Reply 3: I have revised the sentence in lines 48-51 as “We used DXT to analyze the change in structural dynamics of relatively smaller globular proteins, a single-chain variable fragment (scFv) antibody (26 k), a de novo designed protein (8 k), and a DNA-binding protein (12 k), as they interacted with other molecules [12-15].”. I have also added the sentence in lines 325-329 “For small proteins, because structural dynamics could be well determined using NMR such as relaxation and hydrogen-deuterium exchange experiments [2,40], the comparable analyses using both DXT and NMR will provide new insights on the protein structural dynamics and the change upon the ligand binding.”.
Comment 4: The claimed in the title analysis MSD or size of Au nanocrystals is rather ephemeral.
Reply 4: I have added the sentences in lines 321-323 “Using the smaller nanocrystals, it would be hard to get a signal from them. This is one of the critical issues to improve the DXT methods, especially applying for small proteins.”.
Comment 5: Some of the figures have been previously published e,g, Figure 2a = Fig.1 from https://doi.org/10.1016/j.bpc.2021.106661 just the underline removed. citation missing in the captions
Reply 5: I have added the reference [13].
Comment 6: A lot of abbreviations have been introduced without Fv domains, Fv antibodies Fv antibodies, Bcr
Reply 6: I have added the words, variable fragment (Fv) and B cell receptors (BCRs).
Comment 7: Line 82 It was previously shown. please add reference(s)
Reply 7: I have added the reference [21].
Comment 8: What is C6? (G4S)3 linker? HA line 155?
Reply 8: C6 is an affinity maturated and Gly95-type antibody, as described in line 91. (G4S)3 linker is three repeats of the Gly-Gly-Gly-Gly-Ser peptide, as added in line 96. HA is one of the peptides and its amino acid sequence is shown in Fig. 2A.

Reviewer 2 Report
This paper is a mini-review of a specific area of study - the change in protein fluctuations that occurs when a ligand is bound - using the DXT technique developed by a group to which the author belongs. Although the title refers to DXT, ITC was used in all the studies, sometimes in combination with DXT and sometimes without (but perhaps with other biochemical techniques). When both DXT and ITC were used, the results were in agreement, confirming the validity of DXT.
The 3 cases chosen for inclusion represent a good range of ligand binding cases (binding a small molecule, a metal ion, and a DNA oligomer). Results are reasonable and quantify the binding energy in these systems.
A small part of the discussion centers on the issue of relative sizes of the proteins used (8-26 kDa, 5-10 nm diameter) and the gold nanoparticles attached to them for DXT (40-80 nm diameter). Figure 4 shows that fluctuations are suppressed in some cases when the number of protein molecules bound to a nanoparticle is high. The text says that there are two peaks in the angular displacement histogram in this case, although the histogram itself shows only a broad tail on one side of the peak, not two distinct peaks (and if the curve was fitted using two Gaussians, that's not shown in the figure). The author suggests future experiments using smaller nanoparticles; sounds reasonable but could have been a little more specific - what size? Would it be hard to get a signal from smaller particles?
English is very good. I found a few small things that could be fixed:
line 43: change "called as" to "termed"
line 53: change "studied as" to "described as"
line 55: change "composing of" to "composed of"
line 56: change "transduce" to "transduction of"
line 65: change "the Met" to "a Met"
line 91 to "affinity-maturated" to "an affinity-maturated"
line 102: change "correctly" to "correct"
line 147: change "hydrophobic" to "hydrophobic core"
line 156: add commas after "HA" and "Fig2A)"
line 159: change "to label the gold" to "to label with gold"
Author Response
Thank you very much for your valuable comments.
Comment 1: The author suggests future experiments using smaller nanoparticles; sounds reasonable but could have been a little more specific - what size? Would it be hard to get a signal from smaller particles?
Reply 1: I have added the sentences in lines 321-323 “Using the smaller nanocrystals, it would be hard to get a signal from them. This is one of the critical issues to improve the DXT methods, especially applying for small proteins.”.
Comment 2: I found a few small things that could be fixed:
line 43: change "called as" to "termed"
line 53: change "studied as" to "described as"
line 55: change "composing of" to "composed of"
line 56: change "transduce" to "transduction of"
line 65: change "the Met" to "a Met"
line 91 to "affinity-maturated" to "an affinity-maturated"
line 102: change "correctly" to "correct"
line 147: change "hydrophobic" to "hydrophobic core"
line 156: add commas after "HA" and "Fig2A)"
line 159: change "to label the gold" to "to label with gold"
Reply 2: I have revised them as suggested.

Round 2
Reviewer 1 Report
Formally, the author has considered all the recommendations.